# Maternity Blues: A Narrative Review

**DOI:** 10.3390/jpm13010154

**Published:** 2023-01-13

**Authors:** Valentina Tosto, Margherita Ceccobelli, Emanuela Lucarini, Alfonso Tortorella, Sandro Gerli, Fabio Parazzini, Alessandro Favilli

**Affiliations:** 1Department of Obstetrics and Gynecology, Giannina Gaslini Children’s Hospital, 16147 Genova, Italy; 2Section of Obstetrics and Gynecology, Department of Medicine and Surgery, University of Perugia, 06156 Perugia, Italy; 3Section of Psychiatry, Department of Medicine and Surgery, University of Perugia, 06156 Perugia, Italy; 4Department of Clinical Sciences and Community Health, Medicine and Surgery Faculty, University of Milano, 20122 Milano, Italy

**Keywords:** baby blues, maternity blues, post-partum blues, post-natal blues, puerperium, postnatal mental health

## Abstract

Puerperium is a period of great vulnerability for the woman, associated with intense physical and emotional changes. Maternity blues (MB), also known as baby blues, postnatal blues, or post-partum blues, include low mood and mild, transient, self-limited depressive symptoms, which can be developed in the first days after delivery. However, the correct identification of this condition is difficult because a shared definition and well-established diagnostic tools are not still available. A great heterogenicity has been reported worldwide regarding MB prevalence. Studies described an overall prevalence of 39%, ranging from 13.7% to 76%, according to the cultural and geographical contexts. MB is a well-established risk factor for shifting to more severe post-partum mood disorders, such as post-partum depression and postpartum psychosis. Several risk factors and pathophysiological mechanisms which could provide the foundation of MB have been the object of investigations, but only poor evidence and speculations are available until now. Taking into account its non-negligible prevalence after childbirth, making an early diagnosis of MB is important to provide adequate and prompt support to the mother, which may contribute to avoiding evolutions toward more serious post-partum disorders. In this paper, we aimed to offer an overview of the knowledge available of MB in terms of definitions, diagnosis tools, pathophysiological mechanisms, and all major clinical aspects. Clinicians should know MB and be aware of its potential evolutions in order to offer the most timely and effective evidence-based care.

## 1. Introduction

Puerperium is a period of great vulnerability for the woman, associated with intense physical and emotional involvement. Indeed, changes in postpartum mood are complex and involve biological, psychological, social and cultural components. Maternity blues (MB), also known as baby blues, postnatal blues, or post-partum blues, include low mood and mild, transient, self-limited depressive symptoms, which can be developed in the first days after delivery.

In a recent systematic review and meta-analysis, among included studies, a prevalence of 39%, ranging from 13.7% to 76%, has been reported [1]. Considering that MB is a well-established risk factor for shifting to more severe post-partum mood disorders and its great prevalence after childbirth, making an early diagnosis is important to provide adequate and prompt support to the mother, which may contribute to avoiding evolution toward more serious post-partum disorders [2,3]. Indeed, MB has been shown to constitute a specific risk factor for the occurrence of post-partum depression (PPD), postpartum psychosis (PP) and irrecoverable emotional and cognitive PPD impairment for women and their neonates [4].

The goal of this paper is to review the literature in order to provide an overview of the disease with pathophysiology, clinical characteristics, risk factors and management.

## 2. Methods

This is a narrative review of published data on maternity blues. The review was reported and qualitatively assessed following the SANRA, the Scale for the Assessment of Narrative Review Articles [5].

We performed the research by employing a narrative review method [6,7]. Electronic databases (PubMed, Scopus and Google Scholar) were consulted until 31 December 2021 (without date restriction) for relevant publications in English focusing on but not limited to the use of the keywords stated. Key search terms were: Maternity blues OR Postpartum blues OR Baby blues OR Postnatal blues OR Third-day blues OR Third-day syndrome. The electronic search and the eligibility of the studies were independently assessed by two of the authors (A.F., F.P.).

All studies (experimental and observational) reported in the English language were eligible. Studies were selected if they provided useful information on definitions, pathophysiology, clinical characteristics, risk factors and management of maternity blues. The first selection was based on the title, the second on the abstract, and the third on the full-text article. The bibliography was also analyzed to include articles that could have been missed. The most relevant articles were considered in this narrative review.

Bias across studies, as well as bias and risks related to the source of funding and conflict of interest of authors of the included studies, were assessed. Eventual disagreements were resolved through discussion.

## 3. Definitions

According to the latest scientific evidence, MB, otherwise called and known as “postpartum blues”, “third-day blues”, “third-day syndrome”, “baby blues”, or “postnatal blues” [2,3,8,9], is a transient psychological condition with potential temporary symptoms as brief crying spells, or tearfulness, irritability or emotional lability, sorrow/weeping, unstable mood, insomnia, anxiety, loss of appetite, and poor concentration that occur in the first days after childbirth [1,10,11,12,13,14,15].

Despite the attention focused on this topic during the last decades, an accepted and shared definition of MB does not exist. A summary of available definitions is shown in Table 1. The reason for this wide variety of definitions is dependent on the fact that no reliable and generalized data collection methodology was available [10,16,17,18,19,20,21,22]. Due to this reason, we do not find an allocation of MB in either the Diagnostic and Statistical Manual of Mental Disorders—Fifth Edition—(DSM-5) or the International Classification of Diseases—Tenth Edition—(ICD-10).

First reported by Moloney in 1952, MB was described as a third-day depression following childbirth, characterized by fatigue, despondency, tearfulness and inability to think clearly as clinical features [23]. In the 60s, Hamilton and Yalom defined MB as a transient, benign, mild syndrome occurring post-partum, characterized by sporadic crying, fatigue, irritability and mild confusional state [24,25]. In the following 20 years, psychiatrists agreed on some aspects of MB symptoms, such as frequency and benignity, occurring from about the third to fifth post-partum days and disappearing before the tenth day. They did not agree on whether a depressed mood was typical of the disease [28]. Handley et al. considered depressed mood as characteristic of MB [31], while others disputed this notion, suggesting that some researchers misinterpreted tearfulness as being indicative of depressive mood [25,28].

There is no agreement on which days the blues symptoms reached their peak also. Pitt and Levy reported the peak incidence on the third day post-partum, whereas Davidson found that the greatest risk of the blues was the first 3 days after delivery. A peak of symptoms from the fourth day to the sixth was reported by most of the authors [19,26,27,29,30,32,33,34,35].

Interestingly, Levy demonstrated that MB is not solely related to the puerperium, as a similar type of dysphoria occurs in women following surgery. An important difference between these two conditions is that post-operative dysphoria reaches the peak immediately after the surgery and post-partum dysphoria starts on day 3 or 4 after delivery [32].

## 4. Epidemiology

Due to the lack of a well-established clinical description, diagnostic criteria, differences in the methodology of the studies, and cultural and geographical contexts, there is a great heterogeneity worldwide in reported prevalence rates of MB.

Several reports showed a markedly different prevalence of MB according to the geographical area in which the investigation was conducted: 27.1% in Japan [36], 31.3% in Nigeria [19], 50–80% in Iran [37], 58% in India [3], 70.3% in Korea [38], and 32.7% in Brazil [39].

Interestingly, the lowest prevalence rate among Asian countries was found in Malaysia at 3.5%, and the highest one was 63.3% in Pakistan [38]. Regarding western countries, the reported prevalence rates ranged from 58–67% in the United States, 10–14% in Canada [30], and 55.2% in Europe [20,35].

Recently, a systematic review and meta-analysis reported a prevalence ranging from 13.7% to 76% in 26 included studies, estimating an overall prevalence of 39% (95% confidence interval [32.3, 45.6]; I^2^ = 96.6%). In particular, the Authors highlighted the greatest prevalence of MB among women in Africa at 49.6% [1].

## 5. Neurobiology/Pathophysiology

The neurobiological patterns of MB represent an intriguing topic of investigation since the 80’s. Various hypotheses have been formulated. Overall, several pieces of research showed that MB could be related to the complex deregulation of brain responses in susceptible women to the dramatic changes in hormone levels typical of the peripartum and postpartum period [11]. Heterogeneous and conflicting data were reported. Pathophysiological pathways suggested to explain MB development are summarized and reported in Table 2.

The peak in symptoms typically arises in coincidence with maximal hormonal changes, with the fall of progesterone, lower estradiol and cortisol and the rise of prolactin [28,40,41,42,43]. Nappi et al. studied the role of allopregnanolone, showing that serum allopregnanolone levels were significantly lower in those mothers experiencing MB with respect to euthymic ones. Progesterone levels did not significantly differ between the two groups analyzed [44].

**Table 2 jpm-13-00154-t002:** Neurobiology-pathophysiology of MB.

Hormones, Pathways	Hypothesis
Gonadal, placental steroids, progesterone [40,41,42]	Withdrawal of progesterone (decreased) as a trigger for mood instability; Thetrahydroprogesterone (THP ^a^) and thetrahydrodeoxycorticosterone (THDOC ^b^): decreased levels; Allopregnanolone—decreased.
Estrogen [40,42,45]	Interference with the hypothalamic-pituitary-ovarian axis; Lower levels after birth are associated with vulnerability to MB.
Prolactin [45]	Imbalanced prolactin secretion (increased) is associated with maternal anxiety and mood symptoms (data are conflicting);
Hypothalamic-pituitary-adrenal axis (HPA) [40,42,43,44,46]	Stress reactivity; Loss of HPA ^c^ axis activity is associated with vulnerability to MB; Cortisol levels (plasmatic, salivary and urinary)—conflicting data; Increased ACTH ^d^ levels were observed in MB mothers.
Serotonin system [47,48,49,50]	Decreased serotoninergic activity was suggested; Metabolism of tryptophan widely analyzed (tryptophan availability index, kynurenine to tryptophan ratio); Lower levels of tryptophan in MB mothers.
Gamma-Aminobutyric Acid [51]	Gamma-Aminobutyric Acid—A receptors density: decreased.
Noradrenergic system [52]	Lower levels of noradrenaline and adrenaline.
Humoral factors [53]	Hypothesized beta-endorphin role: decreased levels in MB?
Hypothalamic-pituitary-thyroid axis [54]	Difficult interpretation of post-partum thyroid hormone levels and mood disorders; Possible link between thyroid dysregulation and MB ^c^; Antepartum FT3 ^e^ and FT4 ^f^ levels seem negatively associated with MB; Postpartum TSH ^g^ and reverse FT3 ^e^ levels higher in blues? No association with thyroid antibody levels was reported.

^a^ THP: thetrahydroprogesterone; ^b^ THDOC: thetrahydrodeoxycorticosterone; ^c^ HPA: hypothalamic-pituitary-adrenal; ^d^ ACTH: adrenocorticothropic hormone; ^e^ FT3: free triiodothyronine; ^f^ FT4: free-thyroxine; ^g^ TSH: thyroid stimulating hormone.

The role of GABA-A receptors sensitivity in the maternal brain was also studied. The rapid fall in the levels of tetrahydroprogesterone (THP) and tetrahydrodeoxycorticosterone (THDOC) after delivery, combined with a reduction of the GABA-A receptors density, may produce a withdrawal syndrome with typical symptoms of the MB [51].

The experience of the MB could be related to the increased secretion of hypothalamic adrenocorticotrophic hormone (ACTH) secretagogue peptides due to the negative-feedback inhibition on the maternal hypothalamus by the fall of placental corticotrophin-releasing hormone (CRH) [46].

Another interesting hypothesis proposed in the genesis of MB is represented by elevated monoamine oxidase levels or decreased serotoninergic activity. The role of serotonin (5-HT), through the study of the metabolism of tryptophan, a precursor of serotonin synthesis, has been widely analyzed [47,52]. Kohl et al. reported that in women without MB, tryptophan concentration increased within 2 days after birth, whereas it did not change in women with postpartum blues. Moreover, the change in kynurenine to tryptophan ratio, which estimates the degree of tryptophan degradation, was also different between the blues group and the controls at days 0 and 2 [48]. Some authors evaluated the brain tryptophan availability index showing a reduction of this index concomitant to MB; a higher intensity of the blues was directly proportional to the tryptophan concentration [49].

The role of noradrenergic transmission in the onset of MB was also speculated [50]. Kuevi et al. monitored circulating noradrenaline and adrenaline on days 2 to 5 post-partum. They found that women who experienced MB had lower levels of noradrenaline and adrenaline [45].

Humoral factors like β-endorphins may have a role in the onset of blues. Elevated plasma β-endorphin levels have been observed during pregnancy and labor, followed by a rapid fall after delivery [53].

Furthermore, thyroid function could be involved in the genesis of MB: antepartum serum triiodothyronine (FT3) and tetraiodothyronine (FT4) correlated negatively with blues scores in the first week postpartum [54]. Ijuin et al. demonstrated that a low FT3 level and primiparity were significantly correlated with the development of MB [55].

Finally, MB was described as more severe in women with early menarche, a history of premenstrual dysphoric disorders or other menstrual-related mood changes [34,56].

Concerning the psychopathology, MB could demonstrate a mother's vulnerability to mood disorders in the different periods of childbearing age [13,57,58]; indeed, as previously reported, MB has been correlated with premenstrual and pregnancy-related dysphoric disorders as well as neuroticism [35,59,60]. The investigation authored by Reck et al. demonstrated a correlation between MB and mothers who enact passive coping strategies within the couple relationship and in their own mother roles. This correlation, developed during the first week after delivery, could be explained by the psychophysically shocking event that childbirth may represent for the new mother, shaking the continuity of self-perception. The actual reality of the newborn, who in most cases is a concentration of health and potential, represents a challenge for a mother with passive coping, causing a harder time in maintaining psychophysically balance [20].

## 6. Risk Factors

Many studies have been focussed on MB risk factors, but only conflicting data are available in the scientific literature (Table 3). These factors include a history of premenstrual dysphoric disorders and/or menstrual cycle-related mood changes [34], a positive medical history of major depression or dysthymia, a family history of mood disorders, socio-cultural features, economic conditions, and relationship conflicts [3].

From previous studies, socio-economic variables (marital problems, lack of social support) and gravidity status (unplanned pregnancy, parity, mode of conception) revealed a possible role in blues vulnerability [11,25,43,65,66,67]. Insufficient maternal care in childhood, as well as poor family support during pregnancy, were also associated with MB [62]. Faisal-Cury et al. observed an approximately four times lower prevalence of experiencing MB among legally married women and non-smokers [39].

Most of the first studies examined the relationship between psychiatric disorder, affective disorder and MB, with conflicting results [35]. Wilkie et al. prospectively studied the relationship between sleep deprivation prior to the birth, during labor and in the early puerperium and the subsequent development of the postnatal blues, observing that night-time labor and a history of sleep disruption in the latter stages of pregnancy may have an etiological importance in the development of postnatal blues [68].

Interestingly, Ferber observed the nature of maternal touch in women experiencing MB; this study described the contribution of parity, showing that primiparous mothers with blues avoided all types of touch, whereas multiparous mothers with blues provided firm touch and holding. Thus, maternal touch could be used as a screening tool for the detection of women at risk [12].

In a study that investigated Nigerian women, the main predictors of MB included significant mood change during pregnancy, past admission during the pregnancy, female babies, and single mothers [19].

Recently, Ntaouti et al. reported in a Greek population, a lower number of previous births, fewer years of marriage and the husband’s occupation are associated with MB occurrence [10]. Interestingly, Akbarzadeh et al. investigated the role of breastfeeding training based on the Beliefs, Attitude, Subjective Norms, and Enabling Factors (BASNEF) model on the intensity of postpartum blues, showing a positive effect on maternal knowledge and attitude to breastfeeding and, consequently, a lower intensity of MB. Breastfeeding may improve the quality of the maternal-infant relationship, encouraging a safe attachment at the beginning of infancy and providing a basis for reducing the maternal level of stress via the hormonal pathway (hypothalamic-pituitary-adrenal axis) [41].

A recent study explored the influence of assisted reproductive techniques (ART) and their impact on MB development: the number of previous ART cycles emerged as the strongest predictor, whereas no significant effect was observed for the conceiving method. These results suggest the usefulness of assessing the quality of life (QoL) during pregnancy, considering mothers with previous ART failures with higher vulnerability for MB [71]. On the contrary, another research based on the Italian population did not confirm this evidence on patients undergoing ART [61]. Further investigations are needed to better clarify this topic.

A French paper considered the role of the psychological impact of labor, in particular, the association between the intensity of childbirth pain and the MB. The Authors speculated that MB could be a reaction to stress caused by childbirth pain. Moreover, pain can be felt as a failure for women who prepared themselves for painless labor [64]. Among risk factors, the mode of delivery was also correlated with MB [67]. In a study on 949 Dutch women, instrumental delivery was reported among independent risk factors for rapid cycling mood symptoms, which are considered an important aspect of maternity blues [21]. Gerli et al. summarized the relationship between some variables and the MB prevalence in an Italian population: no statistically significant difference was found in age, nationality and Body Mass Index (BMI) before pregnancy. Interestingly, no significant differences were highlighted in terms of ART cycles or spontaneous pregnancies and the presence of labor analgesia, while MB was significantly more frequent in the case of cesarean section (CS) [61].

Methyldopa, largely used in pregnancy-induced hypertension disorders, was recently analyzed as a possible risk factor for MB; it has been speculated that methyldopa could be responsible for hormone alteration, reduced cerebral blood flow, and neuronal function impairment [69].

Recently, the possible association between *Toxoplasma gondii* infection and postpartum blues in a Chinese population was evaluated. The rationale and the basis of this speculation were that *T. gondii* chronic infection might be related to personality changes and mood disorders. Toxoplasma seroprevalence was evaluated in mothers with MB and in non-affected mothers, but no significant association emerged [70].

Finally, a very recent paper on MB risk factors confirmed as a great variety of factors, both psychological, sociodemographic and obstetrics, may predispose to MB development. This was a cross-sectional study that investigated 227 Croatian mothers: a prevalence of 19.9% MB was reported; anxious attachments style, oxytocin use, lower birth weight, lower resilience and less perceived social support from family resulted as significant risk factors for MB in this selected population [63].

## 7. MB Measurements

The diagnosis of MB is based on numerous methods of assessment that have been developed over time. The most frequent instruments of MB measurement are standardized interviews, questionnaires and/or mood rating scales (Table 4).

### 7.1. The First MB Measurement Methods

Since MB research started in 1962, a lot of scales were designed for its measurements. The first one was delivered by Hamilton and included eight items: fatigue, crying, anxiety, confusion, headache, insomnia, hypochondriasis, and hostility toward partner [24]. Later, in 1973, Pitt made a revision of the Hamilton Scale, adding the item of depression to the already established symptoms: the Blues Rating Scale. The participant herself established the ratings of their symptoms through an interview, but nowadays, no validation of this scale has been reported, mainly because depression is not regarded as a feature of blues [26,30]. Furthermore, in 1968, Yalom et al. suggested a scale including nine items: nervousness, physical discomfort, crying, sadness, mood change, fatigue, irritability, sleep disorder and confusion [25,28].

In 1972, Davidson proposed a subjective anxiety rating scale based on crying episodes and sad days and, according to the number of cries and/or sad days, defined mild or severe blues. Six visual analog mood scales concerned with the symptoms of happiness, depression, tears, anxiety, irritability and lability were developed by Kendell in 1981 [29,30].

In 1980, Handley et al. used a semistructured interview and the Multiple Affective Adjective Checklist to score the women for their overall mood, emotional lability, number of crying episodes, anxiety, insomnia, anorexia and irritability [31].

### 7.2. Stein Maternity Blues Scale

In 1980, Stein developed the Stein Maternity Blues Scale, which included 13 symptoms: depression, crying, anxiety, calmness, restlessness, exhaustion, dreaming, appetite, headache, irritability, poor concentration, forgetfulness and confusion. The scores could range from zero to 26: a score from three to eight designed a mild degree of blues, while a score of nine or more meant severe blues [27,72]. In 1983 Spielberger introduced the State-Trait Anxiety Inventory (STAI) scale to diagnose anxiety and to distinguish it from depressive syndromes [73].

### 7.3. Edinburgh Postnatal Depression Scale

In 1987, Cox, Holden and Sagovsky developed a screening test known as the Edinburgh Postnatal Depression Scale (EPDS), consisting of 10 questions with a score from zero to 30. A score higher than 12 was diagnostic for postpartum depression, and a score from nine to 12 was predictive of it [74,80]. It consisted of demographic characteristics (age, marital status, educational level, obstetric features such as parity or mode of delivery), previous personal or familial history of depression and possible depression triggers that occurred during pregnancy or in the early postpartum. The EPDS is commonly the most widely used scale to assess depressive symptoms in the postpartum period. A score of 13 is used as the cut-off point, analogically to the risk of post-partum depression evaluation. This test is a screening tool, and in case of a positive result, further clinical evaluation is needed. Recently, Zanardo et al. demonstrated that women with higher MB scores represent a distinct subgroup of post-partum women at increased risk of depression symptoms [81].

### 7.4. Kennerley’s Blues Questionnaire

In 1989, Kennerley and Gath devised the Kennerley’s Blues Questionnaire, the only blues tool developed by systematic psychometric methods: it was established after six stages of interviews with mothers divided into different samples. In the first stage, 100 newly delivered mothers were asked to describe unusual emotions experienced in the first days after delivery. In the second stage, a total of 47 words were found and assembled in a questionnaire that was delivered to 100 other newly delivered women. The 32 items resulting from this stage were used for the third stage questionnaire for the other 50 mothers. In the fourth stage, 28 items resulted and were divided into clusters by statistical techniques. The fifth stage was needed to verify the reliability of the 28-item questionnaire by administering it to 112 mothers. Finally, it was tested by administering the same questionnaire to women who were not in the puerperium but were in the first days of minor gynecological procedures. As a result, it consisted of 28 items that have been shown to be valid and reliable. A mother is asked to indicate how she has been feeling that day for each of the 28 items; if an item was present, it was given a score of 1, and if absent, a score of 0. The most frequent items were tearful, tired, anxious, over-emotional, up and down in mood, low-spirited, and muddled in thinking. The non-inclusion of depression suggests that a depressed mood should not be regarded as typical of MB [30,34].

### 7.5. Other Standardized Psychological Scales

Finally, the most recent studies about MB included other psychometric tools to better evaluate the internal women dimension. For instance, the Mieczyslaw Plopa and Jan Rostowsky Marriage Questionnaire (KDM-2) measuring marriage dimensions [78]; the Berlin Social Support Scales (BSSS) evaluating the need for support and the perceived available support for women [77]; the NEO Five-Factor Personality Inventory (NEO-FFI) evaluating neuroticism, extraversion, openness to experience, agreeableness and conscientiousness [2,37,75,76]. Most recently, a study proposed the Item Development of Maternal Blues Suryani (MBS) Scale in the antepartum period that could predict postpartum blues. This scale includes 24 items: internal variables (maternal roles and tasks), eight items and external variables (cultural and social support), 16 items [79].

Moreover, in 2021 a study aimed to develop a maternal blues scale through observation of bonding attachments to predict postpartum blues. The analysis produced 32 items consisting of 24 items regarding the mother’s role and duties as internal factors and eight factors involving social, cultural, and economic support as external factors [82].

Since MB is a transitory condition occurring during the first week or so after delivery with fluctuating symptoms, it is difficult to identify a standardized and universal instrument for the measurement of incidence, severity and duration of the blues. The heterogeneity of the medical expression makes it difficult to detect in clinical practice, but objective and reliable methods of measurement through questionnaires/scales could help to recognize this condition early.

## 8. Management

MB may disturb infant care and increase the risk of symptoms of postpartum depression [81], impair maternal-infant interactions, and affect child development [83]. However, suicidal ideation is not present in postpartum blues, and no specific treatment has been developed [84].

Given the heterogeneous evidence regarding possible etiologies of MB, it is unclear whether prevention strategies would be effective in decreasing the risk of developing this condition. Overall, educating women during pregnancy about postpartum blues may help to prepare them for these symptoms that are often unexpected and concerning in the setting of excitement and anticipation of a new baby [22,80,84].

It is important to reassure new parents that low mood symptoms after childbirth are common and transient. Obstetric providers may recommend that mothers and their families prepare ahead of time to ensure the mother will have adequate support and rest after the delivery. Supportive management seems to be the best strategy in MB condition. The relational dimension of care, including good-quality counseling, talking during the postpartum period, and the accurate observation of emotional responses are critical elements to identify postpartum blues and make a differential diagnosis with others possible and more severe disorders: obstetricians, midwives, and pediatricians should have a high grade of attention [85,86]. Interesting approaches proposed as useful in influencing MB symptoms were music therapy and acupressure. Lee observed that music therapy had a positive effect on decreasing postpartum blues and increasing maternal attachment of puerperal women [87]. A study proposed acupressure on mothers with MB showing a decreased EPSD score, but further data are needed [88]. Recently, an early screening method based on Android applications for baby blues detection was proposed as a promising screening strategy [89].

## 9. Conclusions

Despite relevant knowledge having been collected and several risk factors having been identified regarding MB, there is much to discover yet. Understanding the physiopathological pathway by which MB is developed could lead to new weapons to prevent and face related, more serious, psychiatric disorders.

Even if MB has been generally considered a transient, self-limited, cross-cultural phenomenon, the correct identification of this condition remains difficult because a shared definition and well-established diagnostic criteria are not still available. Certainly, a shared and recognized worldwide definition and an effective tool for diagnosis, easily adaptable to different cultural and geographical contexts, would be of great help for clinicians to better diagnose MB and identify cases at risk of post-partum serious mood disorders, such as post-partum depression and postpartum psychosis. Moreover, shared definitions and diagnostics criteria would lay the basis for good quality and fruitful research. Much regarding MB remains to be known. New evidence has emerged in these years, and others will come out yet [1,61,82,89]. Future research should not be limited to consolidating what has already been highlighted, but it will have to go further by exploring new fields that could bring new knowledge to better understand this complex condition. In this scenario, the role of the father and the risk of “paternity blues”—a transient dysphoric period affected by identifying with the newborn’s vulnerability as well as with the mother’s postpartum vulnerability—which could lead to a shared madness with the mother, could represent an intriguing new aspect to investigate of MB [90].

In conclusion, clinicians should know MB and be aware of its potential evolutions in order to offer the most timely and effective evidence-based care. An appropriate and early diagnosis of MB in clinical settings can give the possibility of assisting mothers with adequate psychological support, making a prompt identification of an eventual shift to the most severe mood disorders. Sensitivity, awareness, and multidisciplinary intervention networks are key elements to guarantee women and their families the best support and preserve the mother-child relationship.

## Figures and Tables

**Table 1 jpm-13-00154-t001:** Maternity Blues: definitions over time.

Authors	Year	Definition	Peak Incidence
Moloney [23]	1952	Third-day depression following childbirth	Third post-partum day
Hamilton [24]	1962	Transient mood disorder occurring in the first days after delivery	From third to fifth post-partum day
Yalom et al. [25]	1968	Transient benign mild depression occurring post-partum (also called post-partum blues syndrome)	Fifth post-partum day.
Pitt [26]	1973	Transitory depression and tearfulness, also known as mother’s blues or third-, fourth-, or tenth-day blues or the transitory syndrome	Third or fourth post-partum day
Stein [27]	1982	Transient and common condition of dysphoria of the early puerperium	Third or fourth post-partum day
Kennerley et al. [28]	1986	Brief and benign affective disorder occurring soon after childbirth	First week after delivery
Kendell et al. [29]	1981	Shortlived lability of mood in the early days after delivery	Fifth post-partum day
Beck [30]	1991	Transitory phenomenon of mood changes consisting of depression symptoms	Within 10 days after delivery
Henshaw et al. [13]	2004	Dysphoric episodes (also called postnatal blues), with symptoms resembling depression but having a shorter duration	From third to fifth post-partum day
Pop et al. [21]	2015	Period of intense short-lasting mood changes, commonly described as ‘the blues’	Third-fifth post-partum day
Ntaouti et al. [10]	2020	Transient physiologic and psychological disorder with potential symptoms of depression and confusion	Third post-partum day

**Table 3 jpm-13-00154-t003:** Risk factors for MB.

Domain	Risk Factors	Association
Gynecological	History of premenstrual dysphoric disorders or other menstrual-related mood changes [34]	+
Medical	Personal history of mood disorders [34,39,61]	++
	Family history of mental disease [3,25,35]	+
	Smoke [39]	+
Psychosocial	Low maternal care in childhood [62]	+
	Lower level of education [3,39,62,63]	++
	Lack of social support [63]	++
	Marital status [39]	+
	Years of marriage [10]	+
	Low economic status [3,11,25]	++
Obstetrical	MB or other mood disorders in previous pregnancy [12,19,41,64]	++
	Parity (conflicting results) [11,25,43,65,66,67]	+/−
	Unplanned versus planned pregnancy [11,25,43,65,66,67]	+
	Sleep disruption [68]	+
	Maternal touch [12]	+/−
	Female baby [19]	+
	Labor course (childbirth pain) [61,64,67]	+
	Mode of conception ART (conflicting results) [61,63]	+/−
	Breastfeeding [37]	+
	Mode of delivery [21,61,67]	+
	History of previous CS [61]	+
	Epidural analgesia [19,37,64]	−
	Birth weight (lower) [63]	+
	Previous voluntary interruption of pregnancy [61]	+
	Alpha-metildopa use [69]	+
	Toxoplasma Gondii infection [70]	−

Legend: +: positive association; ++: strong positive association; +/−: conflicting data/no solid evidence; −: negative association. ART: assisted reproductive technologies. CS: cesarean section.

**Table 4 jpm-13-00154-t004:** MB measurements.

Scale and Author	Year	Items	Diagnostic Score
Hamilton Scale Hamilton J.A. [24]	1962	Fatigue, crying, anxiety, confusion, headache, insomnia, hypochondriasis and hostility towards partner	>15
Yalom Scale Yalom I. et al. [25]	1968	Nervousness, physical discomfort, crying, sadness, mood change, fatigue, irritability, sleep disorder and confusion	Mean depression score: 120
Davidson Scale Davidson J.R.T. [33]	1972	Number of crying episodes and number of sad days	>2 cries and/or >2 sad days
Blues Rating Scale Pitt B. [26]	1973	Fatigue, crying, anxiety, confusion, headache, insomnia, hypochondriasis, hostility towards partner and depression	0–48
Stein Maternity Blues Scale Stein G.S. [72]	1980	Depression, crying, anxiety, calmness, restlessness, exhaustion, dreaming, appetite, headache, irritability, poor concentration, forgetfulness and confusion	>9
Kendell Scale Kendell et al. [29]	1981	Happiness, depression, tears, anxiety, irritability and lability	>60
State-Trait Anxiety Inventory (STAI) scale Spielberg C.D. [73]	1983	Forty items exploring trait anxiety and state anxiety	>40
Edinburgh Postnatal Depression Scale (EPDS) Cox J., Holden J.M., Sagowsky R. [74]	1987	Ten questions about demographic characteristics, previous personal or familiar history of depression and possible depression triggers during pregnancy/early postpartum	>13
Kennerley’s Blues Questionnaire Kennerley H., Gath D. [34]	1989	Twenty-eight items include tearful, tired, anxious, over-emotional, up and down in mood, low-spirited, and muddled in thinking	No score
NEO Five-Factor Personality Inventory (NEO-FFI) Zawadzki, B.; Strelau, J. Widiger, T.A.; Costa Jr., P.T. [75,76]	1998	Neuroticism, extraversion, openness to experience, agreeableness and conscientiousness	No score
Berlin Social Support Scales (BSSS) Schwarzer, R.; Schulz, U. [77]	2003	Perceived available support, need for support, support seeking currently received support, and protective-buffering scale and subscales (emotional support and informational support)	Arithmetical mean or a scale or subscale in the range of 1–4
Mieczyslaw Plopa and Jan Rostowsky Marriage Questionnaire (KDM-2) Popla M.; Rostowsky J. [78]	2008	Intimacy, similarity, self-realization, disappointment, and the total score (global assessment of relationship/marriage)	No score
Development of Maternal Blues Suryani (MBS) Scale Suryani M. et al. [79]	2019	24 items: - internal variables (maternal roles and tasks), 8 items - external variables (cultural, social support), 16 items	>15 for internal variables >33 for external variables

## Data Availability

Not applicable.

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
