# Peer review of "Maternity Blues: A Narrative Review"

_jpm, 2023, doi:10.3390/jpm13010154_

Round 1

Reviewer 1 Report

The authors provide an overview of the definition, epidemiology, pathophysiology, risk factors, measurement and management of the maternity blues. This is a prevalent condition that requires better understanding and further research is needed to advance its recognition and management.

The authors have presented a summary of the research on maternity blues covering a broad range of issues. A broad overview such as this is likely to be useful to busy clinicians and the effort to include different components in the overview is to be commended. However, while the authors have not set out to conduct a systematic or structured literature review, this overview is not sufficiently detailed to evaluate how comprehensive it is. For example, there is no description of the strategy used to identify relevant studies or explanation of how the research was selected for inclusion. There is little reference to the quality of the included studies, for example whether they were sufficiently large or adequately powered. There is little attempt to quantify the strength of the effect sizes for risk factors, for example what does 'strong positive association' mean? Therefore, the overview needs further detail, both on the methods used and the research included.

Given the lack of a universally accepted definition of maternity blues and limited understanding of what factors increase women's risk after childbirth, a systematic review to summarise all of the existing evidence focusing specifically on this component (or perhaps one of the other components) would be an excellent addition to the literature. 

Reviewer 2 Report

In the present study, authors present an interesting overview of different aspects concerning one of the most frequent mood disorders during postpartum, such as the so-called “baby blues” also called “postpartum blues” and “maternity blues”.

Because it is a usually self-limited medical condition, it has not received sufficient attention in the research field. Therefore, there are no standardized criteria for its definition, diagnosis, and treatment and much remains to be known. For this reason, I consider it is a very interesting topic.

This is a generally well-written and interesting manuscript. However, I recommend minor corrections and clarifications to improve the paper presentation.

keywords: most are synonyms. I suggest replacing one of them with postnatal mental health

Manuscript

Line 57 …making an early diagnosis it is…. delete “it”

Line 58 … which may contribute to avoid evolutions… replace evolutions for evolution

Line 73 … both DSM-5 and ICD-10. I recommend introducing acronyms with full terminology the first time you use them

Lines 80, 114, 127,165, 179, 196, 253, 259, 277 When using the term “et al.” please include a full stop after.

Line 171 typo mistake “an etiological”

Line 198 BMI. I recommend introducing acronyms with full terminology

Table III the + - signs are misaligned in the Obstetrical domains

Finally, although the literature on whether fathers also experience postpartum blues is lacking, given that few studies  showed that prenatal and postpartum depression affects about 10% of men, it might be interesting to include some comments about postpartum blues in fathers.
